# Alveolar Ridge Augmentation with a Novel Combination of 3D-Printed Scaffolds and Adipose-Derived Mesenchymal Stem Cells—A Pilot Study in Pigs

**DOI:** 10.3390/biomedicines11082274

**Published:** 2023-08-16

**Authors:** Chau Sang Lau, Jasper Chua, Somasundaram Prasadh, Jing Lim, Leonardo Saigo, Bee Tin Goh

**Affiliations:** 1National Dental Research Institute Singapore, National Dental Centre Singapore, Singapore 168938, Singapore; gmslauc@nus.edu.sg (C.S.L.); leonardo.saigo@singhealth.com.sg (L.S.); 2Oral Health Academic Clinical Programme, Duke-NUS Medical School, Singapore 169857, Singapore; 3Duke-NUS Medical School, Singapore 169857, Singapore; jasper.chua@u.duke.nus.edu; 4Center for Clean Energy Engineering, University of Connecticut, Storrs, CT 06269, USA; somasundaram.prasadh@uconn.edu; 5Osteopore International Pte Ltd., Singapore 618305, Singapore; lim_jing@osteopore.com

**Keywords:** alveolar ridge augmentation, 3D-printed scaffolds, adipose-derived mesenchymal stem cells, tissue engineering, pig model

## Abstract

Alveolar ridge augmentation is an important dental procedure to increase the volume of bone tissue in the alveolar ridge before the installation of a dental implant. To meet the high demand for bone grafts for alveolar ridge augmentation and to overcome the limitations of autogenous bone, allografts, and xenografts, researchers are developing bone grafts from synthetic materials using novel fabrication techniques such as 3D printing. To improve the clinical performance of synthetic bone grafts, stem cells with osteogenic differentiation capability can be loaded into the grafts. In this pilot study, we propose a novel bone graft which combines a 3D-printed polycaprolactone–tricalcium phosphate (PCL-TCP) scaffold with adipose-derived mesenchymal stem cells (AD-MSCs) that can be harvested, processed and implanted within the alveolar ridge augmentation surgery. We evaluated the novel bone graft in a porcine lateral alveolar defect model. Radiographic analysis revealed that the addition of AD-MSCs to the PCL-TCP scaffold improved the bone volume in the defect from 18.6% to 28.7% after 3 months of healing. Histological analysis showed the presence of AD-MSCs in the PCL-TCP scaffold led to better formation of new bone and less likelihood of fibrous encapsulation of the scaffold. Our pilot study demonstrated that the loading of AD-MSCs improved the bone regeneration capability of PCL-TCP scaffolds, and our novel bone graft is suitable for alveolar ridge augmentation.

## 1. Introduction

Alveolar ridge augmentation is a dental surgical procedure performed on patients with tooth loss to maximize the amount of bone tissue required to support a dental implant [1]. Under natural conditions following a tooth extraction, the alveolar bone around the empty socket would undergo resorption and the alveolar ridge would recede in height and width [2]. Besides leading to aesthetic and functional issues, the reduced bone volume would also cause insufficient support to the dental implant, causing the implant to become loose and unstable [3]. For patients facing such issues, alveolar ridge augmentation is necessary to regenerate the bone tissue and restore the alveolar ridge so as to ensure the long-term stability of the implant.

To increase the volume of bone tissue in a resorbed ridge, augmentation is often carried out with the help of bone grafts [4]. The current gold standard for bone grafting is the use of autogenous bone or bone isolated from another location in the same patient, such as the mandibular ramus, iliac crest and calvarium. However, the harvest of autogenous bone subjects the patient to additional surgeries, morbidities, costs and recovery time [5]. Bone grafts can also be harvested from another patient or another animal species [6]. Such grafts, known as allografts and xenografts, spare the patient from the harvest of bone from another part of the body, but they introduce risks of disease transmission and adverse immune reactions [7]. They also have batch variability, and the use of porcine or bovine xenografts may be incompatible with certain religions [8].

Driven by the high demand for bone grafts worldwide, not only for alveolar ridge augmentation but also for other types of bone regeneration, researchers are developing novel bone grafts from various biological materials [9,10] and synthetic materials [11,12] to overcome the limitations of naturally derived bone grafts. One synthetic material that is increasingly used for bone grafting is polycaprolactone (PCL), a biocompatible, non-toxic and bioresorbable polymer [13]. PCL is the main component in a number of clinical products approved by the US Food and Drug Administration (FDA), such as Capronor™, an implantable subdermal contraceptive device, and Monocryl™, a resorbable monofilament suture [14,15]. PCL is also the main component in the bioresorbable scaffolds fabricated by Osteopore International, a company specializing in three-dimensional (3D)-printed scaffolds for bone regeneration. Due to its low melting point (60 °C), PCL can be 3D-printed into scaffolds with customizable shapes and structures to optimize cell attachment, while the similarity in mechanical property between PCL and bone and the slow degradation kinetics of PCL make PCL effective in supporting, protecting and maintaining the space in a bone defect [16]. Using a patented fused deposition modeling (FDM) technology, Osteopore produces a number of FDA-approved clinical PCL scaffolds, such as the OsteoPlug™ for the repair of neurosurgical burr holes and the OsteoMesh™ and OsteoStrip™ for orbital floor reconstruction [17,18,19,20]. Osteopore International has also developed a composite scaffold consisting of 80% PCL and 20% tricalcium phosphate (TCP), a bioactive mineral similar to the hydroxyapatite found in bone. The biological performance of PCL scaffolds improved after the addition of 20% TCP [21] and the PCL-TCP scaffold showed enhanced mechanical and biochemical properties compared to the first generation of PCL scaffolds [22,23].

The efficacy of Osteopore’s PCL and PCL-TCP scaffolds for bone regeneration has been reported in a number of studies [19,20]. Our group previously reported a pilot clinical trial where the use of a PCL scaffold for ridge preservation resulted in statistically significantly less vertical ridge resorption and higher bone volume compared to the control group where no scaffold was used [24]. Our group also performed another study that investigated the use of a PCL-TCP scaffold for bone regeneration in a monkey tooth socket facial wall defect model [25]. In this monkey study, the test group with the PCL-TCP scaffold showed better maintenance of the alveolar contour when compared to the control group with autogenous particulate bone at 6 months. However, while bone ingrowth was observed in the area where the scaffold was in contact with the bony socket wall, a lack of bone regeneration was observed on the labial side of the scaffold, where a bony wall is absent. Upon further investigation, the lack of bone regeneration was attributed to a proliferation of fibroblasts from the labial soft-tissue flap [25]. In another study, PCL endoprostheses implanted into monkey mandibular segmental defects contributed to better bone regeneration than the control group, with no scaffolds after 6 months, but the PCL group still showed incomplete bone union [26]. These observations suggest that PCL and PCL-TCP scaffolds still exhibit limitations in oral–maxillofacial bone regeneration.

Due to the complex anatomy surrounding the oral cavity and the multidirectional forces faced by oral–maxillofacial bone tissues during jaw movement, it is more challenging to regenerate bone in the oral–maxillofacial region than in other parts of the body [27,28]. To overcome the challenges in oral–maxillofacial bone regeneration and to improve the clinical performance of the existing PCL-TCP scaffolds, we propose a novel solution that combines a 3D-printed PCL-TCP scaffold with adipose-derived mesenchymal stem cells (AD-MSCs) that are harvested, processed and implanted within the alveolar ridge augmentation surgery.

AD-MSCs have been found to be capable of differentiating into all mesogenic lines, including chondrogenic, fibromuscular, osteogenic and adipogenic cell lines [29]. They have significant advantages over the more commonly used bone marrow-derived mesenchymal stem cells, including: (1) a higher concentration of cells in adipose extracts than bone marrow extracts, (2) larger quantities of aspirate available, (3) less invasive harvest method and (4) less donor site morbidity [30,31]. The adipose tissue can be harvested via a closed-syringe harvesting technique and can be processed rapidly to yield a purified, AD-MSC-rich graft to be loaded into the 3D-printed scaffold for implantation [32,33]. We hypothesized that the addition of AD-MSCs to the scaffold would improve the bone formation in the defect area compared to the scaffold without cells. To test our hypothesis, we performed a small-scale pilot study using a porcine lateral alveolar defect model and compared the performance of the novel scaffold–cell combination with the performance of the PCL-TCP scaffold alone.

## 2. Materials and Methods

### 2.1. Scaffold

A polymer mineral composite consisting of 80% PCL and 20% TCP was 3D-printed via FDM into 8 × 8 × 3 mm scaffold blocks by Osteopore International Pte Ltd., Singapore. The PCL-TCP filaments were printed in a three-angle layering (0°/60°/120°) format with a horizontal gap of 2 mm and a vertical gap of 1 mm. The scaffold blocks were individually packaged, vacuum-sealed and sterilized with gamma radiation.

### 2.2. Animals

Five male adult domesticated pigs (*Sus scrofa domesticus*) with complete adult dentition were obtained for this study. All animal procedures were conducted in the SingHealth Experimental Medicine Centre (SEMC) according to the guidelines of the American Association for Accreditation of Laboratory Animal Care (AAALAC) and approved by the Institutional Animal Care and Use Committee of Singapore Health Services (under protocol 2020/SHS/1560).

### 2.3. Equipment and Surgical Items

The AdiPrep system from Terumo BCT, Inc. (Lakewood, CO, USA) was used for the extraction and processing of adipose tissue. The AdiPrep system consisted of a centrifuge (SmartPrep 2) and a gamma-sterilized disposable set (ADI-25-01), which included cannulas for infiltration and lipoaspiration, as well as a closed syringe system for the harvesting and transferring of adipose tissue.

Instruments and gowns for the surgery were sterilized via autoclave and cooled to room temperature before use. Consumables for the surgery, such as scalpel blades, needles, syringes and sutures, were obtained presterilized.

### 2.4. Surgery

The surgical procedure was performed according to our previously reported protocol [34] and is illustrated in Figure 1. In summary, the procedure consisted of two surgeries: (i) a tooth extraction surgery and (ii) an implantation surgery, which was performed 3 months after the first surgery. The animals were given soft diets for 4 weeks before each surgery and were fasted overnight before each surgery to prevent regurgitation and vomiting during anesthesia. The animal was sedated with an intramuscular injection of ketamine, atropine and xylazine and placed under general anesthesia with 5% isoflurane for induction and 2–3% for maintenance.

In the tooth extraction surgery, the bilateral mandibular mucoperiosteal flaps from the first premolars to the first molar were raised and the four premolars were extracted by sectioning the roots to ensure preservation of the cortical bone. Primary closure was performed with sutures and the extraction sites were left for 3 months to induce bone resorption.

In the implantation surgery, the animal was placed in a supine position after anesthesia and 30 mL of lipoaspirate was extracted from the abdominal subcutaneous adipose tissue. The animal was then turned over to the prone position and the bilateral mucoperiosteal flaps were raised again at the previous extraction sites. Four defects of 8 × 8 mm (two on the left and two on the right) were created on the buccal side of the alveolar ridge using drill burs. During the raising of the flaps and creation of the defects, the lipoaspirate was processed with the AdiPrep system to yield 2 mL of stromal vascular fraction (SVF), a concentrated cellular extract of adipose tissue containing a heterogeneous population of AD-MSCs, stromal cells and immune cells. The 8 × 8 × 3 mm PCL-TCP block was opened from its sterile packaging and the SVF was loaded into the pores of the porous scaffold block via a 21 G needle. Each of the 4 defects was randomly selected to receive one of the following: (1) scaffold with cells, (2) scaffold without cells, (3) autologous bone (harvested from the drilling of the other defects), and (4) no graft. For the scaffold with cells, scaffold without cells, and autologous bone groups, the implant was secured to the buccal cortex with a titanium microscrew (Product number 04.503.608.01C, DePuy Synthes, Raynham, MA, USA) to prevent movement of the scaffold. After implantation, each defect was covered with a collagen membrane (BioGide, Geistlich, Wolhusen, Switzerland) and the periosteum was closed using sutures. The extraction of the adipose tissue and the preparation of the cell-loaded scaffold are illustrated in Figure 2.

The vital signs of the animal were constantly monitored by the veterinarian team during surgery and post-operation recovery. The animal was only returned to its pen when swallowing, palpebral/corneal reflexes were present. The surgical sites were examined 1 week, 1 month and 2 months after surgery with the animal under sedation to check for adverse events such as infection or wound dehiscence. The animal was kept on a soft diet and enrichment toys made from hard materials were removed from the pen for the entire duration of the study to prevent the animals from exerting mechanical stress on the operated alveolar ridge.

Three months after the implantation surgery, the animal was euthanized via a perfusion fixation method where 300–500 mL of Hartman’s solution and 800 mL of a mixture of 2.5% paraformaldehyde and 2% glutaraldehyde were perfused into the left ventricle. The mandible was separated from the rest of the body and the left and right mandibular segments containing the surgical sites were harvested. The specimens were fixed in 10% paraformaldehyde for 3 days and stored in 70% ethanol. All procedures were performed at room temperature.

### 2.5. Micro-Computed Tomography and Histology

Each specimen was subjected to micro-computed tomography (µCT) scanning using the nanoScan^®^ for single-photon emission computed tomography/computed tomography (SPECT/CT) (Mediso, Budapest, Hungary) with the following parameters: number of projections = 2; scan method = semicircular multi-field of view (FOV); X-ray power = 70 kVp, 280 µA; exposure time = 170 ms. The micro-CT data were analyzed using the software VGSTUDIO MAX 3.5 (Volume Graphics, Heidelberg, Germany). The threshold values for background, ISO value and material were set at −1000, 2600 and 12,000 Hounsfield units (HUs), respectively. The region of interest was defined as a rectangular block with the same dimensions as the scaffold (8 × 8 × 3 mm), with the titanium microscrew passing through the center of the block. The bone volume fraction and bone density in each region of interest were calculated and recorded.

After µCT measurements, the specimens were dehydrated progressively in 70%, 80%, 95% and 100% ethanol for 1 day each, followed by infiltration in xylene, methyl methacrylate (MMA), and 95% MMA/5% dibutyl phthalate (DBP) for 2 days each (with 1 h of vacuum degassing at the start of each infiltration step). The specimens were embedded in 95% MMA/5% DBP/0.2% Perkadox-16 at room temperature. The MMA-embedded specimens were sectioned with a diamond-blade saw (EXAKT 300 CP Band Saw, EXAKT, Norderstedt, Germany). Each section was attached to an acrylic slide with a clear adhesive and polished progressively with P320, P800, P1200 and P2400 sandpaper on a rotary grinder (EXAKT 400 CS Micro Grinding System, EXAKT, Norderstedt, Germany) until the section was around 50 µm thick. The polished sections were stained with methylene blue and basic fuchsin and scanned at 10x magnification with the TissueFAXS slide scanner (Tissuegnostics, Wien, Austria). The exported images were analyzed using Adobe Photoshop 22.3.1 (Adobe, San Jose, CA, USA). Descriptive analysis was performed on the histology images to evaluate the bone regeneration and any inflammatory reaction, while histomorphometry measurements were carried out for histological grading.

### 2.6. Statistical Analysis

As the sample size of this study was small, the analysis of new bone formation and histomorphometric parameters was performed mostly in a descriptive fashion. For the µCT measurements, simple statistical analysis was performed in the form of a Student’s *t*-test to compare the bone volume fraction between the treatment groups, with *p*-values of <0.05 considered statistically significant.

## 3. Results

### 3.1. Clinical Observations

The tooth extraction surgeries of the five animals were carried out smoothly and all five recovered well after the surgery. The alveolar ridge of the five animals demonstrated evident reduction in width and height over 3 months. The implantation surgeries of the five animals were also carried out successfully without hiccups. However, the first animal exhibited wound dehiscence, scaffold exposure and significant inflammation at the surgery site one week after the implantation surgery. As the surgery sites had been compromised, the decision was made to omit the first animal from subsequent analysis. The other four animals remained stable and healthy throughout the study, with no adverse complications observed at the 1-week post-surgery check, 1-month post-surgery check and 2-month post-surgery check. All five pigs were euthanized and their mandibular segments harvested 3 months after the surgery, but only the mandibular segments of the four healthy pigs were subjected to downstream analysis.

### 3.2. Visual Analysis of 3D Model

Representative 3D µCT images of the mandibular segments were reconstructed digitally to evaluate the formation of new bone at the surgical sites (Figure 3).

From the reconstructed images, the defect sites belonging to the “scaffold only,” the “scaffold with cells” and the “autologous bone” groups were observed to be occupied with bone tissue, with the head of the titanium screw remaining visible above the bone ingrowth. The defect receiving nothing (belonging to the “blank” group) had also healed, with the surgical defect becoming indistinguishable from the original bone. The presence of bone ingrowth in the “scaffold only,” the “scaffold with cells” and the “autologous bone” groups, as well as the bone healing in the “blank” group, were consistently observed in all four animals.

### 3.3. Quantitative µCT Analysis

The bone volume percentage (BV%) represents the amount of mineralized bone formed in the defect space after 3 months of healing time. The BV% values, expressed as 100% × (bone volume/total volume), are presented for each individual animal in Figure 4a and presented as the mean and standard deviation in Figure 4b.

The “autologous bone” group had the highest BV% in all four animals, with the values being consistent across the four animals (pig 2 = 30.0%, pig 3 = 49.9%, pig 4 = 45.2%, pig 5 = 52.3%). On average, the “autologous bone” group had a BV% of 43.7%. On the other hand, the PCL-TCP scaffold alone (the “scaffold only” group) had a BV% of only 18.6% on average, as the BV% value of the “scaffold only” group was low in the four animals (pig 2 = 10.5%, pig 3 = 34.8%, pig 4 = 15.2%, pig 5 = 14.0%). When AD-MSCs were loaded into the PCL-TCP scaffolds, the BV% improved by 10% to 28.7% on average. Improvement in BV% upon addition of AD-MSCs to the PCL-TCP scaffold was observed in all four animals, and the trend was consistent (BV% value for “scaffold with cells” group: pig 2 = 19.6%, pig 3 = 42.3%, pig 4 = 27.0%, pig 5 = 25.7%)

As the defects in the “blank” group healed well naturally, the defect became indistinguishable from the surrounding original bone. The difficulty in establishing the boundary between the defect and the original bone made the BV% values for the “blank” group highly variable and inaccurate. Therefore, the BV% values for the “blank” group are not reported in this study.

To determine the efficacy of the PCL-TCP scaffold and the PCL-TCP scaffold loaded with AD-MSCs compared to the autologous bone, the efficacy ratio for the “scaffold only” or “scaffold with cells” treatment group was computed for each animal using the following equation, and the results are shown in Table 1:$$\mathrm{Efficacy}\;\mathrm{ratio}=\frac{\mathrm{BV}\%\;\mathrm{of}\;“\mathrm{scaffold}\;\mathrm{only}”\;\mathrm{or}\;“\mathrm{scaffold}\;\mathrm{with}\;\mathrm{cells}”}{\mathrm{BV}\%\;\mathrm{of}\;“\mathrm{autologous}\;\mathrm{bone}”}\;$$

As seen from Table 1, when compared to the use of autologous bone, the use of PCL-TCP scaffold alone is about 41% effective, while the addition of AD-MSCs to the PCL-TCP scaffold improve the efficacy of the treatment by 24%, making the scaffold–cell combination about 65% effective. In the individual animals, the improvement in efficacy after adding AD-MSCs to the scaffold ranges from 15% (in pig 3) to 30% (in pig 2).

### 3.4. Descriptive Histology

A total of 16 specimens (four treatment groups × four animals) were subjected to histological evaluation. For the “scaffold only”, “scaffold with cells” and “autologous bone” groups, the titanium microscrew was used as a landmark to identify the center of the defect. The section bisecting the central axis of the microscrew was taken as the middle section. Three sections—the middle section, the section before it and the section after it—were used for the histological analysis. For the “blank” group, due to the absence of a titanium microscrew, three sections spaced 200 µm apart were taken approximately from the central region of the defect. The middle sections of the 16 specimens are shown below in Figure 5. Selected sections are shown at higher magnifications in Figure 6.

In the specimens belonging to the “scaffold only” and “scaffold with cells” groups (Figure 5a–h), the PCL-TCP scaffold was dissolved during histological processing and appeared as empty spaces in the sections. Various amounts of tissue were observed in the spaces between PCL-TCP filaments. In the “scaffold only” group, the spaces around the PCL-TCP filaments were occupied by pockets of soft connective tissue and partially mineralized connective tissue (Figure 5a–d and Figure 6a,b). In some sections, inflamed soft connective tissue was observed (Figure 6c,d). Almost all sections in the “scaffold only” group showed the presence of fibrous encapsulation, visible in the sections as an intensely dark layer surrounding the defect space (Figure 5a–d).

In the “scaffold with cells” group, the spaces around the PCL-TCP filaments were also occupied by pockets of soft connective tissue and partially mineralized connective tissue (Figure 5e–h). In some sections, pockets of mineralized bone matrix interspersed with small bone marrow cavities could be observed in the spaces between the PCL-TCP filaments (Figure 6e–h). Fibrous encapsulation of the defect space was present in pig 2, but was mostly absent in the other three animals (Figure 5e–h).

In the “autologous bone” group, the titanium microscrew was mostly surrounded by dense bone tissue (Figure 5j–l), except in pig 2, where part of the defect was occupied by inflammation (Figure 5i). In pigs 3, 4 and 5, a high amount of bone was observed in the grafted region with no signs of inflammation or bone resorption. Part of the autologous bone was replaced by new bone and the whole region was very compact and rich in osteons, but poor in bone marrow (Figure 5i–l and Figure 6i,j).

In the “blank” group, although the specimens appeared to have healed well upon visual inspection and in the reconstructed µCT model, the histological sections revealed an asymmetrical cross section of the alveolar ridge, with evident resorption on the buccal side of the ridge (Figure 5m–p). While the native bone on the lingual side of the ridge appeared porous with large bone marrow cavities, the bone tissue near the surgical region on the buccal side appeared dense and compact.

Among the four animals, pig 2 had the lowest amount of new bone formation across all treatment groups, demonstrated by the low BV% values from the µCT results. Pig 2 also had the highest level of immune response among the four animals, due to the presence of fibrous encapsulation in the “scaffold only” and “scaffold with cells” histological sections, and the presence of inflamed tissue and incomplete healing in the “autologous bone” histological section. On the other hand, pig 3, with efficacy ratios of 0.70 for the “scaffold only” group and 0.85 for the “scaffold with cells” group, had much better bone formation in the implantation sites than the other three animals after 3 months of healing. Although variability in bone regeneration was observed across the four animals, all showed a consistent trend where the “scaffold with cells” group generally performed better than the “scaffold only” group in µCT and histology.

## 4. Discussion

Bone regeneration is a complex process that starts with the recruitment of osteoprogenitor cells followed by osteogenic differentiation, formation of collagen matrix and mineralization of the matrix [35]. Although bone has a remarkable ability to spontaneously remodel and regenerate after an osseous trauma, bone defects that are critically sized and/or lack a bony wall would not be able to heal spontaneously and would necessitate the use of additional elements such as bone grafts, scaffolds, cells and/or growth factors to achieve satisfactory regeneration [36]. In dentistry, bone regeneration procedures are often carried out in the form of alveolar ridge augmentation, where the aim is to build sufficient bone in the alveolar ridge to enable dental implant placement.

In our study, we developed a novel bone graft for alveolar ridge augmentation and evaluated the efficacy of this bone graft in a porcine lateral alveolar defect model. The bone graft that we developed is a combination of a 3D-printed PCL-TCP scaffold and AD-MSCs obtained autologously in the same session as the alveolar ridge augmentation surgery. The 3D-printed PCL-TCP scaffold used in this study was fabricated by Osteopore International Pte. Ltd., and it demonstrated promising results when used on its own or combined with cells, biological mediators and/or surface treatments in various animal models, including rat [37,38], rabbit [39], dog [40], micropig [41] and monkey [25] (Table 2). In most of these animal studies (including our study), the PCL-TCP scaffold was able to maintain the space of the bone defect and allowed the infiltration of osteogenic cells into the porous scaffold interior. However, when the PCL-TCP scaffold was used on its own without any additives, the amount of new bone formed in the defect was still limited, especially when compared to autologous bone, which is considered the gold standard for bone grafting. In the “scaffold only” group of our study, where the implant was the PCL-TCP scaffold alone, we observed soft connective tissue and partially mineralized connective tissue within the scaffold pores.

One factor that could contribute to the limited bone formation in the “scaffold only” group is the presence of a fibrous layer, which is a sign of fibrous encapsulation, an immune response mechanism to protect the host tissue from the PCL-TCP scaffold, which the body considers a foreign object [42]. The fibrous layer can also be the effect of gingival epithelization, a defensive action by the gingival tissue to rapidly protect the alveolar defect from the oral environment [43]. Be it fibrous encapsulation or gingival epithelization, the presence of rapidly proliferating fibroblasts can hinder the migration and proliferation of osteogenic cells, leading to a lack of osteogenesis. Furthermore, in our lateral alveolar defects, only one side of the scaffold was in contact with a bony wall, while the other sides of the scaffold had limited or no bony contact. In this one-wall unprotected defect, infiltration of osteogenic cells would be limited to the side with bony contact, as observed in a previous pig study [41] and a previous monkey study [25]. This observation was also present in our study, where osteogenic cells and partially mineralized tissue were observed mostly near the bony wall (Figure 6b), while the outer sides of the scaffolds were either occupied with soft connective tissue or unoccupied (Figure 5a,c,d).

To improve the bone regeneration capabilities of PCL-TCP scaffolds, researchers in previous studies have added cells, biological mediators and/or surface treatments, with varying degrees of success (Table 2). One of the most effective ways to enhance the bone regenerative capability of PCL-TCP scaffolds is the addition of mesenchymal stem cells (MSCs), as demonstrated in a previous canine study where the loading of bone marrow mesenchymal stem cells (BM-MSCs) into PCL-TCP scaffolds improved the BV% from 17.3% to 48.6% [40]. MSCs are non-hematopoietic stromal stem cells that have the capacity to regenerate mesenchymal tissue types such as bone, cartilage, ligament, muscle and adipose tissue due to their ability of self-replication and differentiation into multiple mesenchymal lineages [44]. MSCs can be derived from many locations, including bone marrow, periosteum, skin, muscle, tendon, umbilical cord, vessel wall, adipose, and dental tissue, and they play an important role in bone formation and bone regeneration [44].

BM-MSCs have been the traditional choice for bone regeneration, as BM-MSCs were the first MSCs identified and have been extensively studied and characterized [45]. Although very effective in differentiating into bone cells and regenerating bone tissue, BM-MSCs have the limitations of a painful procedure to aspirate bone marrow, low cell yield from bone marrow aspirates, potential complications from the procedure and limited multipotency with increasing passage of the cells or increasing age of the patient [46,47]. The low cell yield of BM-MSCs from bone marrow aspirates also necessitates the expansion of cells in a clean facility to achieve sufficient cell numbers, thus leading to increased time and cost for therapies involving BM-MSCs.

Ever since adipose tissue was identified independently by Zuk [48,49] and Halvorsen [50] as a source of multipotent stem cells capable of differentiating into various cell lines, including cells in the osteogenic lineage, adipose-derived mesenchymal stem cells (AD-MSCs) have been an alternative source of BM-MSCs because of their easy access, higher cell yield and higher proliferation rate [51]. When induced into osteoblasts in vitro, AD-MSCs have been shown to upregulate alkaline phosphatase (ALP) activity, express osteogenic proteins and initiate mineralization in the extracellular matrix (ECM) [48,49].

Many studies were performed to compare the osteogenic potential between BM-MSCs and AD-MSCs, and the results suggested that AD-MSCs seem to be as effective as BM-MSCs in terms of osteogenic capability [45,51,52]. When used together with suitable scaffolds, osteo-differentiated AD-MSCs were able to form osteoid and support bone regeneration in vivo [53,54,55]. When seeded into 3D-printed PCL-TCP scaffolds, AD-MSCs were able to differentiate along the osteogenic lineage in vitro and in subcutaneously implanted scaffolds in athymic rats [56,57]. All these results showed that AD-MSCs are an effective stem cell source for bone regeneration.

In all these reported studies, the isolation of AD-MSCs from adipose tissue consisted of an enzymatic digestion of the adipose tissue with collagenase to obtain a cellular stromal vascular fraction (cSVF) and a cellular extraction containing a variable cell population, including AD-MSCs, fibroblasts, endothelial cells and macrophages [31]. The AD-MSCs were then isolated from the cSVF by centrifugation and/or plastic adherence, and the AD-MSCs were expanded in a culture media supplemented with fetal bovine serum (FBS). For human applications, the FBS used in cell expansion can be replaced by human serum or human platelet lysate. On the other hand, collagenase digestion is often included in the isolation to separate the cells from the network of flexible collagen fibers that forms the adipose tissue [58]. Although collagenase digestion leads to high yield of cells from adipose tissue, the procedure is time-consuming, expensive, and resource-intensive (requiring dedicated equipment and experienced personnel) [58]. Furthermore, the use of collagenase in adipose tissue digestion falls outside the “minimal manipulation” guidelines on autologous implantation set by regulatory agencies [59,60], which dictates that adipose tissue should be minimally manipulated, intended for homologous use, and not combined with other materials [61]. The use of collagenase is considered “more than minimally manipulated,” as collagenase alters the original characteristics of the adipose tissue and can also affect the phenotypical and functional characteristics of the isolated cells [58,62].

To meet the “minimal manipulation” requirements, AD-MSCs can be isolated via enzyme-free methods such as the use of mechanical disruption to break down the extracellular matrix and structural elements of the adipose tissue [63]. After mechanical disruption of the adipose tissue, the adipose extract can be further condensed by centrifugation to remove the oil and aqueous fractions to yield a cellular concentrate, known as the tissue stromal vascular fraction (tSVF) [58]. The tSVF has a variable cell population similar to the cSVF obtained by enzymatic digestion of adipose tissue. The tSVF also contains cellular debris, blood cells, and ECM fragments [64]. One advantage of using tSVF as a whole rather than isolating the AD-MSCs is that the native ECM and perivascular structures present in tSVF provide structural support for the AD-MSCs and help to reduce cell death and improve graft retention [65]. In addition, using tSVF as a whole eliminates the need for cell expansion and allows the extraction, processing and grafting of the adipose cells within one surgery session.

The idea of using adipose tissue and cells for clinical applications is not new, as autologous fat grafting has been performed in cosmetic plastic surgery since the 1980s, albeit with unpredictable outcomes. In the early years, it was believed that the transplantation of the intact mature adipocytes was the most important goal [33]. However, with the discovery of AD-MSCs and their multipotency from the early 2000s, researchers realized the importance of AD-MSCs in autologous fat grafting. Since then, protocols have been developed to obtain cSVF, tSVF and AD-MSCs efficiently from lipoaspirates and the use of adipose tissues and cells started to move beyond cosmetic plastic surgery and into other regenerative applications, with promising results [31]. While there are numerous studies on the use of cSVF and AD-MSCs obtained via collagenase digestion for bone regeneration, there is very little literature on the use of tSVF obtained via mechanical disruption for bone regeneration, especially for oral–maxillofacial applications. Hence, our study serves to fill the gap by evaluating the combination of a 3D-printed PCL-TCP scaffold and tSVF obtained via mechanical disruption for alveolar ridge augmentation.

The results from our study showed that the addition of AD-MSCs in the form of tSVF into the pores of the PCL-TCP scaffold significantly increases the BV% from 18.6% to 28.7% in the porcine lateral alveolar defect model in a span of 3 months. Compared to the “scaffold only” group, the presence of osteogenic cells and mineralized tissue in the “scaffold with cells” group was more evenly distributed across the whole scaffold and no longer limited to the region near the bony wall (Figure 6f,h). In addition, compared to the “scaffold only” group, there were less or no infiltration of soft connective tissue in the outer side of the scaffold in the “scaffold with cells” group. This observation was expected, as the occupation of the scaffold pores by the tSVF should prevent the entry of fibroblast and epithelial cells. On the other hand, the general good health of the four animals after surgery and the lack of complications showed that the combination of the PCL-TCP scaffolds and the implanted autologous adipose tSVF are safe.

Another notable finding of this study is that the addition of AD-MSCs in the form of tSVF to the PCL-TCP scaffolds led to a reduced immune response and a lower incidence of fibrous encapsulation compared to the use of PCL-TCP scaffolds alone. This observation suggests that the presence of immunomodulatory cells in the adipose tSVF may help to modulate the immune system’s reaction to the PCL-TCP scaffold, which may be considered “foreign” by the body. Analysis of the adipose SVF has revealed the presence of multipotent stromal cells that can control the activities of immunomodulatory cells and secrete immunomodulatory cytokines [66]. Such immunomodulatory properties could potentially contribute to improved integration of the scaffold with the host tissue, promoting better overall bone regeneration outcomes.

The researchers acknowledge the variability between animals that was observed in the study. For example, pig 2 showed a lower tissue regeneration across all treatment groups and the presence of inflammation in the “autologous bone” group, while pig 3 exhibited very good tissue regeneration across all treatment groups, including the “scaffold only” group. However, a consistent trend of the “scaffold with cell” group performing better than the “scaffold only” group was observed across all the animals. This consistency strengthens the validity of our findings and indicates that the observed effects are likely not coincidental. However, one limitation of this study is the relatively small sample (n = 4), which limits the generalizability of the results. Also, as this is a simple pilot study, we were only able to have one time point of harvest at 3 months post-surgery. In pigs, a post-surgery duration of 3 months represents mid-term tissue regeneration. Ideally, a duration of 6 months would allow the PCL-TCP scaffold to fully resorb and enable the team to study the long-term tissue regeneration. To validate the findings of this study and to evaluate the robustness of the observed trends, further studies with larger samples and a longer time point are warranted.

## 5. Conclusions

In conclusion, within the limitations of the small sample, we managed to show that the addition of AD-MSCs in the form of tSVF to PCL-TCP scaffolds contributes to better bone formation in a large animal model and that our combination scaffold is safe and suitable for alveolar ridge augmentation. The ability to extract the adipose tissue, process the tissue to obtain the SVF, load the SVF into the scaffold and implant the scaffold in one surgery session without cell expansion will save a lot of time and costs for patients and clinicians.

## Figures and Tables

**Figure 1 biomedicines-11-02274-f001:**
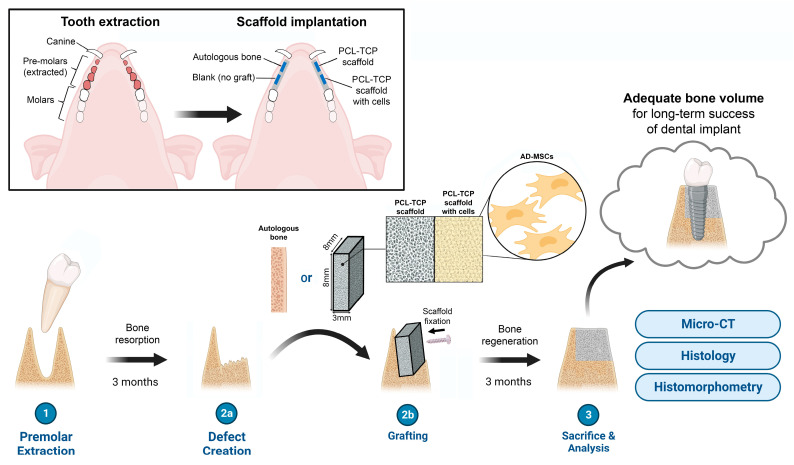
Schematic overview of the study, illustrating the major procedures and the location of the tooth extraction and scaffold implantation in the pig mandible. Adapted from “Dental Implant Procedure in Pigs,” by BioRender.com (2023). Retrieved from https://app.biorender.com/biorender-templates (accessed on 4 April 2023).

**Figure 2 biomedicines-11-02274-f002:**
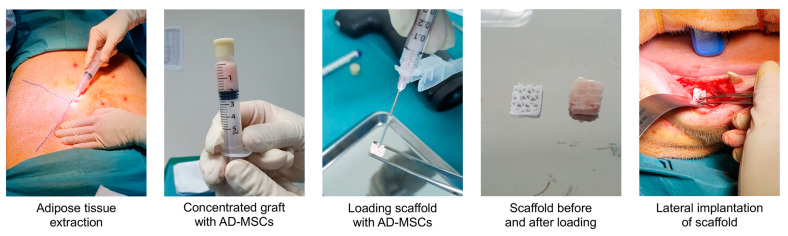
Representative images from the implantation surgery, where the adipose tissue was extracted and concentrated into a stromal vascular fraction rich in AD-MSCs. The injectable SVF graft was loaded into a porous PCL-TCP scaffold via a 21 G needle and the loaded scaffold was implanted laterally on the buccal side of the alveolar ridge. Reproduced with permission from Mary Ann Liebert, Inc. publishers, the publisher of this copyrighted figure.

**Figure 3 biomedicines-11-02274-f003:**
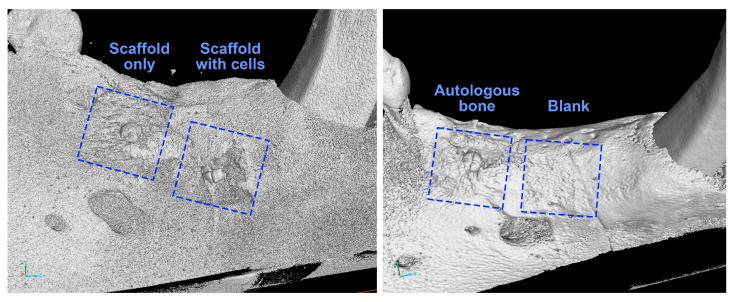
Representative µCT images of mandibular segments harvested 3 months after the implantation surgery, with the defect sites marked in dashed lines. Length of each square = 8 mm (µCT images generated by VGSTUDIO MAX 2023.1 software.).

**Figure 4 biomedicines-11-02274-f004:**
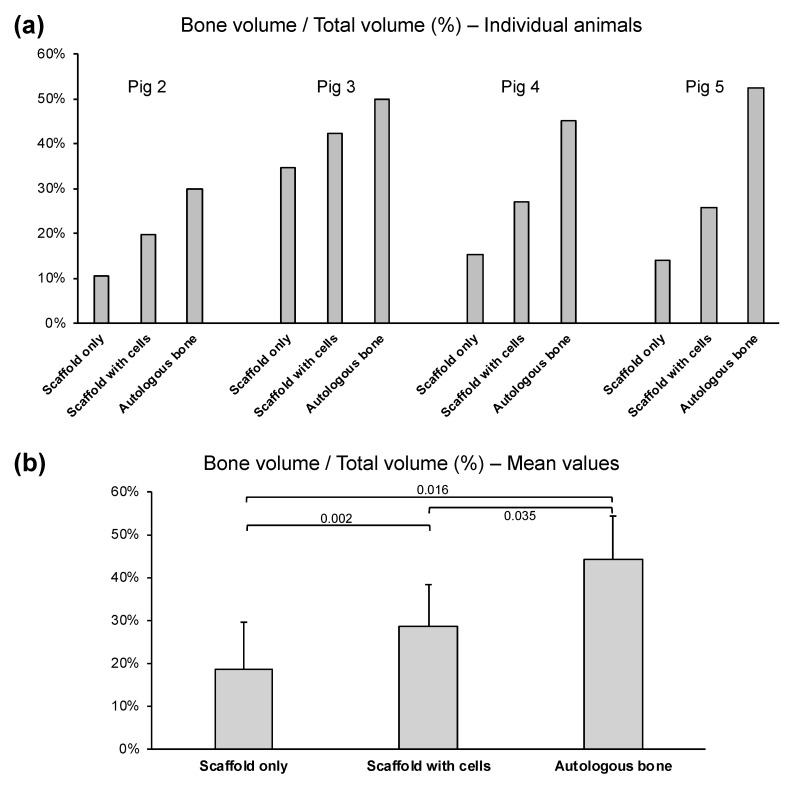
BV% detected 3 months after the implantation surgery. (**a**) BV% of the 3 treatment groups in each of the 4 animals. All 4 animals exhibited a similar trend, where the “autologous bone” group shows the highest BV% and the “scaffold with cells” group has a higher BV% than the “scaffold only” group. (**b**) Mean values of BV% of the 3 treatment groups. Defects treated with autologous bone have the highest BV%, while the addition of adipose cells to the PCL-TCP scaffold significantly improved the BV% (*p* < 0.05). (Scaffold only: mean = 18.6%; scaffold with cells: mean = 28.7%; autologous bone: mean = 43.7%).

**Figure 5 biomedicines-11-02274-f005:**
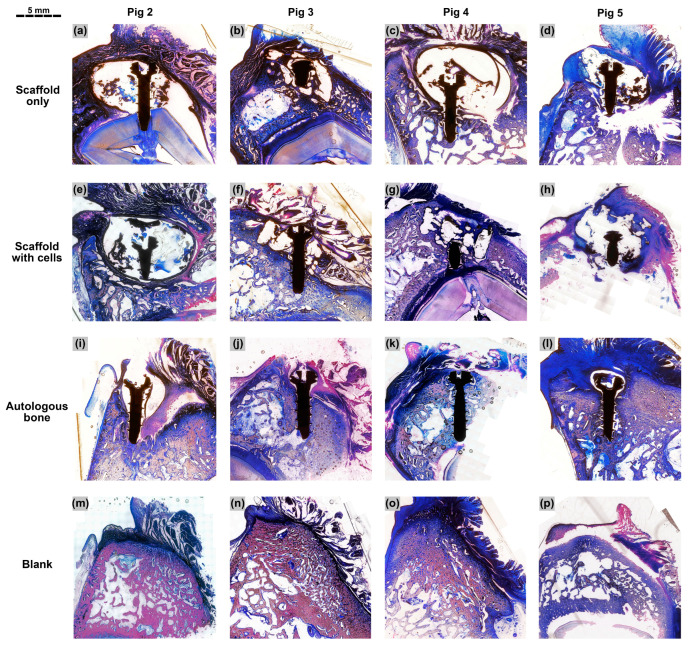
Micrographs of the middle section taken from each of the 16 specimens, stained with methylene blue and basic fuchsin, and shown at 4× magnification. For specimens with titanium microscrews (**a**–**l**), the tip of the screw points towards the lingual side of the alveolar ridge while the head of the screw faces the buccal side. For specimens in the “blank” group (**m**–**p**), the left side is the lingual side of the alveolar ridge and the right side is the buccal side. Scale bar = 5 mm.

**Figure 6 biomedicines-11-02274-f006:**
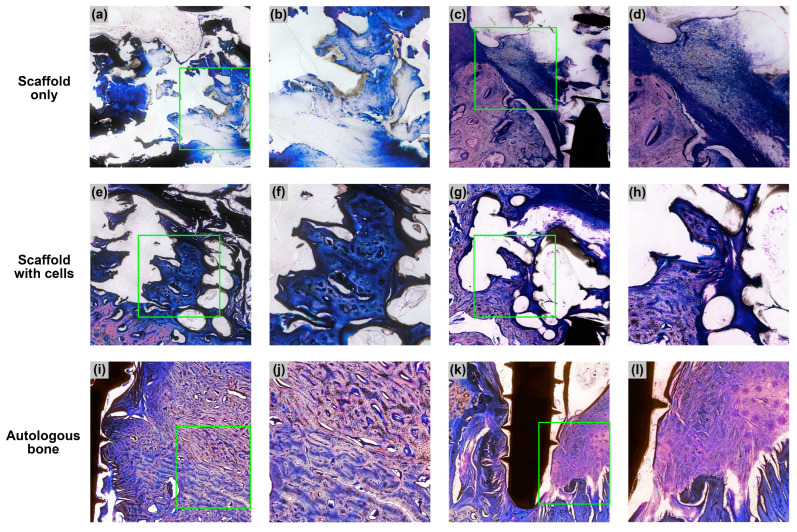
Micrographs of selected sections from the “scaffold only,” “scaffold with cells” and “autologous bone” groups. Images (**a**,**c**,**e**,**g**,**i,k**) are taken at 10× magnification, and the green squares in these images are enlarged to 20× magnification in (**b**,**d**,**f**,**h**,**j**,**l**), respectively. (**a**,**b**): Partially mineralized connective tissues; (**c**,**d**): inflamed soft connective tissue; (**e**,**f**): mineralized bone matrix with small bone marrow cavities; (**g**,**h**): pockets of mineralized bone matrix between the PCL-TCP filaments; (**i**,**j**): boundary between original bone (below) and autologous bone (above); (**k**,**l**): newly formed bone on the left side of the screw and inflamed soft tissue on the right side of the screw.

**Table 1 biomedicines-11-02274-t001:** Efficacy ratio for each individual animal at 3 months after implantation surgery, and the mean and standard deviation of the efficacy ratio for all 4 animals.

Animal	Efficacy Ratio	
	Scaffold Only	Scaffold with Cells
Pig 2	0.35	0.65
Pig 3	0.70	0.85
Pig 4	0.34	0.60
Pig 5	0.27	0.49
Mean ± standard deviation	0.41 ± 0.19	0.65 ± 0.15

**Table 2 biomedicines-11-02274-t002:** Bone volume fraction results of this study and selected animal studies involving PCL-TCP scaffolds fabricated by Osteopore International.

Animal	Surgery Location	Study Duration	Sample	BV%	Reference
Pig	Alveolar ridge	3 months	PCL-TCP only	18.6%	This study
			PCL-TCP + ADMSC	28.7%	
			Autologous bone	43.7%	
Rat	Femur	12 weeks	PCL-TCP only	4.2%	[37]
			PCL-TCP + PRP *	4.6%	
Rat	Calvaria	12 weeks	PCL-TCP–fibrin	23.7%	[38]
			PCL-TCP–fibrin-HS3 *	38.6%	
Rabbit	Calvaria	12 weeks	PCL-TCP only	18.3 mm^3^ **	[39]
			NaOH-treated PCL-TCP *	21.5 mm^3^ **	
Dog	Mandible	8 weeks	PCL-TCP only	17.3%	[40]
			PCL-TCP + BMMSC *	48.6%	
Micropig	Alveolar ridge	6 months	PCL-TCP only	18.0%	[41]
			Autologous bone	51.5%	
Monkey	Mandible	6 months	PCL-TCP only	6.8% ***	[25]
			Autologous bone	11.8% ***	

* PRP = platelet-rich plasma, HS3 = heparan sulfate 3, NaOH = Sodium hydroxide, BMMSC = bone marrow mesenchymal stem cells. ** Values are reported as bone volume (mm^3^). Total volume is not given. *** Values are reported as bone area percentage from histomorphometric analysis.

## Data Availability

The data presented in this study are available on request from the corresponding author. The data are not publicly available due to the presence of copyrighted material from Mary Ann Liebert, Inc. publishers.

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
