# Peer review of "Alveolar Ridge Augmentation with a Novel Combination of 3D-Printed Scaffolds and Adipose-Derived Mesenchymal Stem Cells—A Pilot Study in Pigs"

_biomedicines, 2023, doi:10.3390/biomedicines11082274_

Round 1
Reviewer 1 Report
the introduction must be improved about the bio materials in the past (for example 10.3390/app12031480 and other) and about the new studies about the socket resorbs and the role of periostal Inhibition (10.3390/app122312292 and other). This is in my opinion what you have to improve in a good manuscript.
Author Response
Point 1: the introduction must be improved about the bio materials in the past (for example 10.3390/app12031480 and other) and about the new studies about the socket resorbs and the role of periostal Inhibition (10.3390/app122312292 and other). This is in my opinion what you have to improve in a good manuscript.
Response 1: We thank the reviewer for his/her comments and helpful suggestions. Based on the reviewer's first suggestion, we have added a few references (including 10.3390/app12031480) in line 55 and 56 of the manuscript to give readers a better overview of the biological materials (e.g. decelluarized dentin) and synthetic materials used in bone regeneration. (Please see line 55 and 56 of the attachment)
As for the reviewer's second suggestion, we feel that the role of periostal Inhibition may not be relevant to our study because periostal inhibition is a preventive measure (to stop buccal bone resorption) but our study is focused on the augmentation of alveolar ridge which is already resorbed. (We leave our pigs for 3 months after tooth extraction to simulate patients who comes to the clinic with resorbed alveolar ridge.) If the treatment surgery is carried out immediately after tooth extraction, then the role of periostal Inhibition is more relevant/important. Nevertheless we thank the reviewer for his/her valuable suggestion.

Reviewer 2 Report
Dear authors. Thank you for submitting your study for consideration. I found the subject and methods very interesting.
Firstly, there is a high degree of novelty in the approach described. The data describing the significantly increased amount of bone augmentation compared to the common approach has both research and clinical ramifications. Secondly, the scientific method is sound and highly relevant and reliable, even though the sample size is small, since the split-mouth method was used. Finally, the manuscript is well written
I concluded that this manuscript adds to our body of knowledge on this topic and provides direction for further future research.
Author Response
Point 1:
Dear authors. Thank you for submitting your study for consideration. I found the subject and methods very interesting.
Firstly, there is a high degree of novelty in the approach described. The data describing the significantly increased amount of bone augmentation compared to the common approach has both research and clinical ramifications. Secondly, the scientific method is sound and highly relevant and reliable, even though the sample size is small, since the split-mouth method was used. Finally, the manuscript is well written
I concluded that this manuscript adds to our body of knowledge on this topic and provides direction for further future research.
Response 1: We thank the reviewer for his/her encouraging comments and we also thank the reviewer for his/her valuable time to review our work.